# Operative and Clinical Outcomes of Minimally Invasive Living-Donor Surgery on Uterus Transplantation: A Literature Review

**DOI:** 10.3390/jcm10020349

**Published:** 2021-01-18

**Authors:** Yusuke Matoba, Iori Kisu, Kouji Banno, Daisuke Aoki

**Affiliations:** Department of Obstetrics and Gynecology, Keio University School of Medicine 35 Shinanomachi, Shinjku-Ku, Tokyo 160-8582, Japan; y.matoba0212@gmail.com (Y.M.); kbanno@z7.keio.jp (K.B.); aoki@z7.keio.jp (D.A.)

**Keywords:** uterus transplantation, living donor surgery, laparotomy, laparoscopy, robot assisted, uterine vein, ovarian vein, utero-ovarian vein

## Abstract

Background: The surgical approach and choice of drainage veins for uterus transplantation living-donor surgery have been investigated to reduce invasiveness. Methods: A thorough search of the PubMed database was conducted. The search was not limited by language or date of publication. The data were collected on 13 October 2020. Two reviewers independently assessed each article and determined eligibility for inclusion in the review article. Inclusion criteria were English peer-reviewed articles reporting surgical information or postoperative course, articles regarding animal research on UTx, UTx on deceased donors, or not original articles. Results: Of the 51 operations within 26 articles reviewed, the mean operative time was shortest in the laparoscopic approach, and longest in the robot-assisted approach. The mean blood loss was less in the laparoscopic and robot-assisted approaches than in the open approach. In cases where the uterine veins were not preserved, the mean operative time was shortened by each approach and the mean blood loss decreased with the laparoscopic and robot-assisted approaches. Conclusions: These procedures may contribute to less invasive living-donor surgery.

## 1. Introduction

Absolute uterine factor infertility (AUFI) includes congenital uterine malformation and defects, such as Mayer-Rokitansky-Küster-Hauser (MRKH) syndrome [1], which occurs in one in 5000 women; acquired uterine defects caused by treatment of uterine cancers or hysterectomy due to puerperal bleeding; extended uterine myomatosis; and Asherman’s syndrome, in which the endometrium is adhered [2].

A new transplantation technique, uterine transplantation (UTx), has been clinically applied in recent years for the treatment of AUFI. UTx was first performed in Saudi Arabia in 2000 [3]. Although the world’s first UTx failed with the removal of a transplanted uterus, basic research using animal models was continued, and in 2014, a Swedish team reported the first live birth after UTx [4]. Since then, UTx has been applied clinically in many countries, and there have been some reports of live births from women who have undergone UTx [5].

However, there are medical, ethical, and social challenges to UTx. One of the medical challenges is the highly invasive procedure for living donors. In UTx living-donor surgery, the uterine artery is usually used for the arterial vessel, but there are several venous options. The uterine vein (UV), a branch of the internal iliac vein, is widely used [6], as by the Swedish team that obtained the first live birth after UTx. When the UV is used, the surgical operation is similar to radical hysterectomy. As the surgical isolation of the UV is performed in a narrow and deep area of the pelvis and there is a complex network of vessels, the procedure is sometimes difficult, resulting in longer surgical time and massive hemorrhage. In addition, as the procedure is performed near the hypogastric nerve, there is a risk of postoperative complications such as dysuria in the living donor [7].

To solve this problem, the use of ovarian veins (OV) and utero-ovarian veins (UOV) as drainage veins has been investigated (Figure 1) [8]. When these veins are used, the surgical technique is easier because the vessels to be preserved are in a more superficial layer than when the UV is preserved. In addition, UTx living-donor surgery was initially performed using an open approach, but recently there have been reports of laparoscopic [9] and robot-assisted approaches [10] for donor surgery.

In many cases of UTx performed to date, the uterine veins from the internal iliac vein are used as drainage veins. However, this surgery is challenging because these vessels are located in the deep pelvic floor and surround the ureter. To minimize the invasiveness of living-donor surgery, the use of the ovarian vein or the utero-ovarian vein—which runs continuously from the ovarian vein through the mesosalpinx—as the drainage vein, has been considered as an alternative to the use of the uterine vein (Ut, uterus; UV, uterine vein; UOV, utero-ovarian vein; OV, ovarian vein).

In this review of the literature, we report on the differences in surgical and clinical outcomes by the variation of surgical approach and the preserved veins in UTx living-donor surgery.

## 2. Materials and Methods

### 2.1. Search Strategy

A thorough search of the PubMed database was conducted. The search was not limited by language or date of publication. The search strategies were as follows: (uterus [Title/Abstract] OR uterine [Title/Abstract] OR womb [Title/Abstract]) AND (transplantation OR transplant) AND (“surgery” [Title/Abstract] OR operation [Title/Abstract] OR laparoscopy [Title/Abstract] OR laparoscopic [Title/Abstract] OR robot [Title/Abstract] OR robotic [Title/Abstract] OR laparotomy [Title/Abstract] OR vein [Title/Abstract] OR veins [Title/Abstract] OR venous [Title/Abstract] OR anastomosis [Title/Abstract] OR ovarian [Title/Abstract] OR utero-ovarian [Title/Abstract] OR utero-ovarian [Title/Abstract] OR living [Title/Abstract] OR donor [Title/Abstract] OR livebirth [Title/Abstract] OR live-birth [Title/Abstract] OR human). The data were collected on 13 October 2020.

### 2.2. Eligibility Assessment

Two reviewers (Y.M. and I.K.) independently assessed each article and determined eligibility for inclusion in the review article. Inclusion criteria were English peer-reviewed articles reporting one of the following: (i) surgical information (operative approach, surgical time, blood loss, types and numbers of veins, and operative complications); or (ii) postoperative course (discharge timing, graft failure, and live birth after UTx). Articles regarding animal research on UTx, UTx on deceased donors, not original articles (video article, review, letter to the editor, commentary, and editorial), not written in English, or that did not report the information above were excluded.

### 2.3. Data Extraction and Analysis

The included studies were reviewed by two independent reviewers (Y.M. and I.K.), and relevant data were extracted including the number of performed human UTx cases, surgical approach of living-donor surgery (open approach, laparoscopic approach, or robot-assisted approach), surgical time, blood loss during donor surgery, the types and numbers of removed veins (UV, UOV, or OV), operative complications, discharge timing, and live birth after UTx.

The data were classified into open, laparoscopic, and robot-assisted approaches for analysis.

The data were also classified and analyzed according to whether the UV was removed within each approach.

## 3. Results

This review included 26 original articles (Figure 2). Reports of living-donor uterus transplants from Saudi Arabia [3], Sweden [4,6,11,12,13,14,15,16,17,18,19], China [10,20], USA (Dallas) [8,21,22,23,24], Czech Republic [7,25,26], Germany [27,28], and India [9,29] were identified, and 51 living-donor UTx were incorporated. The surgical information and clinical data for each case are shown in Table 1. In one case in Germany, the uterus was removed from a donor, but was found to be unsuitable for transplantation during back table processing, and the transplant was not performed. In another case, the uterine veins were not used for transplantation, even though they were preserved and removed from the donor in a Czech case.

On 13 October 2020, an article search was conducted on PubMed according to the search strategy. Of 2382 articles, 26 original articles were finally included in the review. They include the operative and clinical outcome data of the UTx living donor. UTx, uterine transplantation

Of the 51 living-donor UTx cases, the open approach was used in 33 cases, the laparoscopic approach in four cases, and the robot-assisted approach in 14 cases. The data of each approach are summarized in Table 2. The average operative time was 8 h 26 min ± 2 h 47 min for the open approach, 3 h 30 min ± 0 h 33 min for the laparoscopic approach, and 10 h 59 min ± 1 h 45 min for the robot-assisted approach, with a trend toward shorter operative times for the laparoscopic approach and longer operative times for the robot-assisted approach. The mean blood loss was 715 ± 584 mL with the open approach, 100 ± 0 mL with the laparoscopic approach, and 209 ± 182 mL with the robot-assisted approach, with a trend toward less blood loss with minimally invasive procedures, such as the laparoscopic and robot-assisted approaches. The day of discharge was 6.2 ± 1.3 postoperative days on average with the open approach, 6.5 ± 0.5 days postoperatively with the laparoscopic approach, and 4.3 ± 1.0 days postoperatively with the robot-assisted approach. There were 19 surgical complications with the open approach (57.6%), zero with the laparoscopic approach (0.0%), and six with the robot-assisted approach (42.9%). There were nine cases (28.1%) of graft failure in open approach, zero cases (0.0%) on the laparoscopic approach, and two cases (14.3%) in the robot-assisted approach. Live birth after living-donor UTx was reported in 16 cases (48.5%) with the open approach, zero cases (0.0%) with the laparoscopic approach, and two cases (14.3%) with the robot-assisted approach.

Clinical data for each operative approach with or without the uterine veins are also shown in Table 2. In the open approach, the mean operative time was 8 h 45 min ± 2 h 39 min and the mean blood loss was 711 ± 586 mL in the cases where UVs were preserved (*n* = 29), and in the cases where UVs were not preserved (*n* = 4), the mean operative time was 6 h 14 min ± 0 h 26 min, and the mean blood loss was 738 ± 569 mL. In the laparoscopic approach, the mean operative time was 4 h 0 min ± 0 h 0 min and the mean blood loss was 100 ± 0 mL in the UVs preserved cases (*n* = 2), and the mean operative time was 3 h 0 min ± 0 h 20 min and the mean blood loss was 100 ± 0 mL in the non-UVs preserved cases (*n* = 2). In the robot-assisted approach, the mean operative time and mean blood loss were 11 h 19 min ± 1 h 8 min and 245 ± 197 mL in the UVs preserved cases (*n* = 12), respectively, and the mean operative time was 9 h 1 min ± 3 h 1 min and the mean blood loss was 100 ± 0 mL in the non-UVs preserved cases (*n* = 2). In each approach, the operative time was reduced in the non-UVs preserved cases. The discharge time was 6.3 ± 1.4 postoperative days for the open approach in the UVs preserved cases and 5.8 ± 0.8 days in the non-UVs preserved cases, and was 7.0 ± 0.0 postoperative days for the laparoscopic approach in the UVs preserved cases and 6.0 ± 0.0 postoperative days in the non-UVs preserved cases. In the robot-assisted approach, the postoperative discharge time was 4.4 ± 1.0 days in the UVs preserved cases and 4.0 ± 0.0 days in the non-UVs preserved cases. There was little difference between patients with and without UVs preserved. Operative complications were found in 17 (58.6%) cases for the open approach with UVs preserved, and in two (50.0%) cases for non-UVs preserved. No complications were reported with the laparoscopic approach in both of the UVs preserved and non-UVs preserved cases. Complications tended to occur more frequently in the robot-assisted approach, with six cases (50.0%) observed solely in the UVs preserved cases, with none in the non-UVs preserved cases. Complications were more frequent in the UVs preserved cases. In the robot-assisted approach, graft failure was reported in two patients (16.7%) with UVs preserved. Live births after UTx utilizing the laparoscopic approach were not reported in any of the papers included in this review. In the robot-assisted approach, one case (8.3%) of a live birth was reported from the UVs preserved cases, and one (50.0%) was reported from the non-UVs preserved cases.

## 4. Discussion

Laparoscopic and robot-assisted approaches may contribute to less invasive living-donor surgery in UTx in the same way that they have led to less invasive surgery for other diseases. Similarly, the use of the OV or UOV for drainage, without UV, may also contribute to less invasive living-donor surgery.

Although UTx is an option for the treatment of AUFI, there are still some issues to be solved. UTx is a non-vital organ transplantation and can be considered a transplantation to improve quality of life, but it requires a lot of surgical burden to the donors and the recipients. There are at least four operations in the UTx: uterus explantation in the donor, uterus implantation in the infertile woman, caesarean section after pregnancy with the aid of assisted reproductive technologies, and hysterectomy after completed family planning. Surgery for living donors takes a long time and causes a large amount of blood loss, which places a heavy burden on the living donor. Surgical invasion of the recipient may be acceptable since infertility is seen as a disease that causes a great psychological burden. However, the living donor is a completely healthy person. The burden for living donors of UTx must be reduced as much as possible.

One of the reasons for the high invasiveness of donor surgery is the difficulty in handling the complex venous system around the uterus. As the veins around the uterus make a network surrounding the ureter, they are difficult to isolate in the narrow operative field of the pelvic floor without sustaining damage. As a solution to this challenge, techniques in which the OV and/or UOV are used, instead of the UV, as drainage veins have been developed. In the UTx cases included in this review, only the UOV and OV were used, instead of the UV in each approach of living-donor surgery. With the exception of the open approach, the blood loss tended to be lower in non-UV preserved cases than in the UV preserved cases. This suggests that not using the UV may contribute to a less invasive living-donor surgery for UTx.

On the other hand, there are concerns about the use of the UOV and OV as drainage veins. One of the concerns is whether these veins are sufficient for blood flow in the gestational uterus. Many pregnancies and live births have been reported after radical trachelectomy (RT), a fertility preservation treatment for cervical cancer in which the venous blood flow of the uterus is dependent on the UOV and OV, as well as UTx, using the UOV and OV only [30]. Considering this fact, the venous flow required by the gestational uterus is preserved even if only the UOV and OV are used as drainage veins. In addition, considering that there have already been reports of live births in UTx without using the UV as a drainage vein [8,20], the use of the UOV and OV should be considered as an option for living-donor surgery. In the case of the use of the OV as a venous vessel on the premenopausal donor, where ovariectomy is inevitable, should be discussed. In an ASRM statement on UTx, it is described that it is generally not recommended to use the ovarian vessels if this results in the loss of ovaries in premenopausal women [31].

Laparoscopic and robot-assisted approaches have advantages, such as improved magnification and small, minimally invasive wounds. These characteristics are being applied to clinical practice in UTx with the possibility of contributing to less invasive living-donor surgery. Previous reports comparing laparoscopic and open approaches for radical hysterectomy, in which the uterine artery and veins are handled as well as UTx living-donor surgery, have reported less intraoperative blood loss with the laparoscopic approach than with open approach [32]. In addition, the robot-assisted approach has already been used in other organ transplant surgeries [33]. It has already contributed to minimizing invasive living-donor surgery and is, therefore, considered feasible. In this review, the laparoscopic and robot-assisted living-donor UTx operations also tended to reduce blood loss compared to the open approach. There was also a tendency for patients to be discharged earlier with the robot-assisted approach than with the open approach. These findings suggest that the laparoscopic and robot-assisted approaches may contribute to less invasive UTx in the future.

On the other hand, the disadvantages and complications of laparoscopic and robot-assisted approaches need to be considered. Fourteen cases of robot-assisted surgeries were reported, but the average operative time was longer than that of the open approach. Robotic-assisted procedure has not been standardized yet, and the robot-assisted approach differs depending on each center. Therefore, it is not possible to clearly state why robotic-assisted approach takes longer than the open approach at the moment. Temporary alopecia has been reported as a complication related to intraoperative positioning in robot-assisted approach [24]. In addition, the main goal of UTx is to achieve live birth after transplantation. Although many live births have been reported for the open approach, partly because of the small number of procedures performed, only two live births have been reported for the laparoscopic and robot-assisted approach UTx. A long-term report on mental status after UTx [17] found no clear results on how live birth affects mental status, but it is thought that live birth could affect the mental status of living donors. Although minimally invasive techniques for living donors are required, these need to be developed with the objectives of UTx in mind.

The open approach has a higher incidence of surgical complications and graft failure than minimally invasive approaches. However, the open approach has been performed since the inception of UTx, and surgical outcomes from that time are also included in this review. In addition, the robot-assisted approach is utilized in countries that perform more UTx by the open approach by surgeons who have sufficient experience with UTx surgery. The differences in clinical and operative outcomes between each approach may be influenced by these factors, which do not lead directly to the conclusion that the open approach is inferior to minimally invasive approaches. Further accumulation of data that regards the clinical outcome of each approach is essential.

UTx from deceased donors has been clinically applied as a fundamental solution to the risks of living donors, and there are reports of live births after UTx from deceased donors [34,35]. To investigate UTx from a deceased donor, basic research has also been conducted [36]. However, this technique has challenges such as hormone replacement therapy for postmenopausal donors and difficulties in assessing the detailed uterine vasculature. The effects of these factors on graft implantation and live birth need to be examined. There are also many issues that need to be considered, such as what criteria are suitable for deceased donors for UTx and what surgical procedures for organ harvesting, such as hysterectomy, may be appropriate.

Limitations are present in this review. As this is a retrospective review, various confounding factors may be included, and it is not possible to say directly which approach is superior. Statistical comparisons with laparoscopic and robot-assisted approaches are particularly difficult to conduct because of the small number of procedures performed. In addition, this review only included original articles that met the inclusion criteria retrieved by PubMed. However, there are some articles that include unpublished data, such as live birth after laparoscopic UTx in India, and some press releases about the implementation of UTx in other countries [5,37,38]. If such information is included, the results may differ from those of this review.

This literature review on UTx living-donor surgical approaches, preserved veins, and operative and clinical outcomes was conducted with a focus on the laparoscopic approach and robot-assisted approach. Laparoscopic and robot-assisted approaches may contribute to less invasive living-donor surgery in UTx in the same way that they have led to less invasive surgery for other diseases. Similarly, the use of the OV or UOV as a drainage vein, without UV, may also contribute to less invasive living-donor surgery. However, there are few reports of live birth—the ultimate goal of UTx—with these new techniques. The application of these techniques needs to be thoroughly discussed.

## Figures and Tables

**Figure 1 jcm-10-00349-f001:**
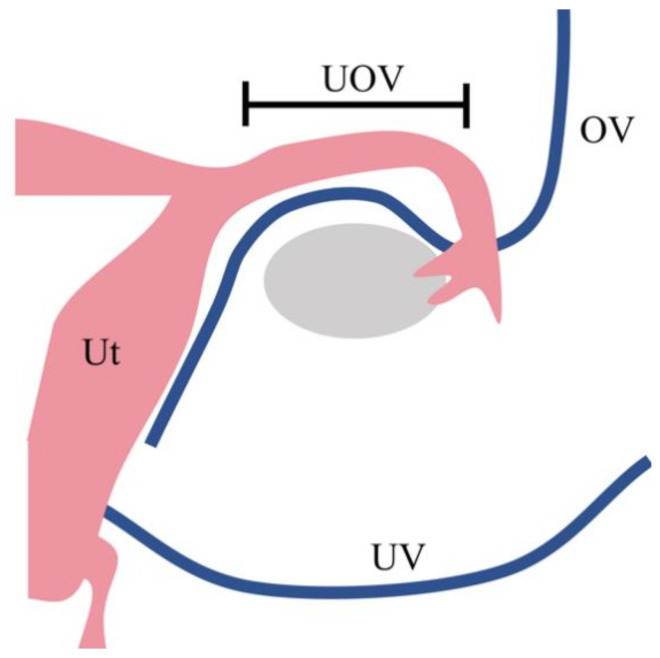
Drainage vein options for uterus transplantation. In many cases of uterus transplantation performed to date, the uterine veins from the internal iliac vein are used as drainage veins. However, this surgery is challenging because these vessels are located in the deep pelvic floor and surround the ureter. To minimise the invasiveness of living-donor surgery, the use of the ovarian vein or the utero-ovarian vein—which runs continuously from the ovarian vein through the mesosalpinx—as the drainage vein, has been considered as an alternative to the use of the uterine vein. Ut, uterus; UV, uterine vein; UOV, utero-ovarian vein; OV, ovarian vein

**Figure 2 jcm-10-00349-f002:**
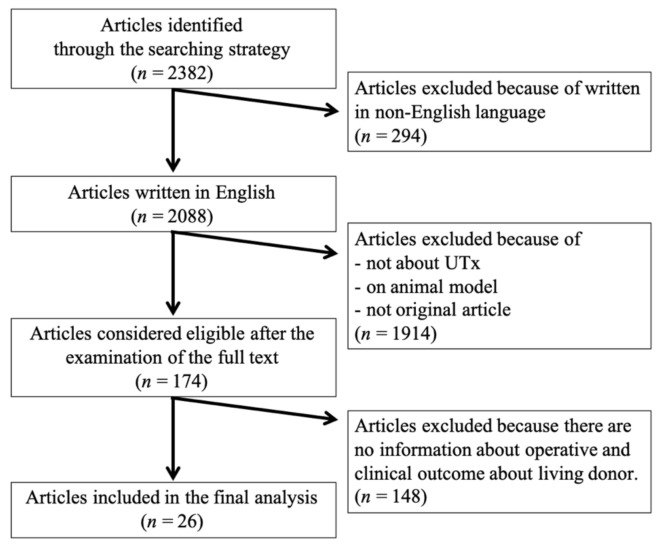
Flowchart of article selection.

**Table 1 jcm-10-00349-t001:** Reported operative and clinical data of living-donor surgery for uterus transplantation.

Country	Operation	No.	Surgical Time (h:min)	Blood Loss (mL)	Preserved Vein	Graft Failure	Operative Complications (Grade *)	Discharge	Live Birth	Remarks
Saudi Arabia [3]	OPEN	1	N/R	N/R	2 × UV	Yes	Intraoperative ureteric injury (N/R)	N/R	N/A	
Sweden [4,6,11,12,13,14,15,16,17,18,19]	OPEN	1	10:54	300	2 × UV, 1 × UOV	No	Nocturia (1)	6POD	Yes × 2	
	OPEN	2	12:37	2400	2 × UV, 1 × UOV	Yes	Wound infection (2) Uterovaginal fistula (3b)	6POD	N/A	
	OPEN	3	12:53	800	2 × UV, 1 × UOV	No	None	6POD	no	
	OPEN	4	10:34	600	2 × UV, 1 × UOV	No	Unilateral sensibility impairment of the thigh (1)	6POD	Yes × 2	
	OPEN	5	10:17	600	2 × UV	No	None	6POD	Yes × 1	
	OPEN	6	10:52	700	2 × UV, 1 × UOV	No	None	6POD	Yes × 2	
	OPEN	7	10:17	400	2 × UV, 1 × UOV	No	None	6POD	Yes × 1	
	OPEN	8	11:23	400	2 × UV	No	None	6POD	Yes × 1	
	OPEN	9	13:08	2100	2 × UV	Yes	None	6POD	N/A	
	ROBOT	1	13:00	600	2 × UV, 1 × UOV	No	None	N/R	N/R	
	ROBOT	2	12:30	400	2 × UV, 1 × UOV	No	Gluteal light pain when walking (N/R)	5POD	Yes	
	ROBOT	3	11:30	N/R	2 × UV, 2 × UOV	Yes	N/R	N/R	N/A	
	ROBOT	4	12:30	N/R	2 × UV, 1 × UOV	No	Pressure alopecia (2)	N/R	N/R	
	ROBOT	5	11:30	N/R	2 × UV, 2 × UOV	No	N/R	N/R	N/R	
	ROBOT	6	11:30	N/R	2 × UV, 2 × UOV	No	N/R	N/R	N/R	
	ROBOT	7	11:30	N/R	2 × UV, 2 × UOV	No	N/R	N/R	N/R	
	ROBOT	8	10:00	N/R	2 × UV, 1 × UOV	Yes	Pyelonephritis (3b)	N/R	N/A	
China [10,20]	ROBOT	1	6:00	100	2 × OV	No	None	5POD	Yes	
US (Dallas) [8,21,22,23,24]	OPEN	1	5:45	400	1 × UV, 1 × UOV	Yes	Leg/buttocks pain (1)	6POD	N/A	
	OPEN	2	7:21	1000	1 × UV, 1 × UOV	Yes	UTI (2)	6POD	N/A	
	OPEN	3	6:41	1300	1 × UV, 1 × UOV	Yes	Vaginal cuff dehiscence (3b) Depression (2), UTI (2)	6POD	N/A	
	OPEN	4	6:40	1700	2 × UOV	No	UTI (2)	5POD	Yes	
	OPEN	5	6:34	250	2 × UOV	No	Faecal impaction (3b)	7POD	Yes	
	OPEN	6	7:07	1100	1 × UV, 1 × UOV	No	Acute blood loss anaemia (2)	5POD	Yes	
	OPEN	7	6:38	600	2 × UV	No	UTI (2)	5POD	Yes	
	OPEN	8	6:12	400	2 × UOV	Yes	None	6POD	N/A	
	OPEN	9	7:34	750	1 × UV, 1 × UOV	No	Symptomatic anaemia (2), UTI (2)	5POD	Yes	
	OPEN	10	6:27	1500	2 × UV	No	Acute blood loss anaemia (4a) Prolonged intubation (4a), UTI (2)	8POD	N/R	
	OPEN	11	5:33	600	2 × UOV	No	None	5POD	Yes	
	OPEN	12	5:13	950	1 × UV, 2 × UOV	Yes	Haemorrhage (N/R)	4POD	N/A	
	OPEN	13	6:10	800	1 × UV, 1 × UOV	No	UTI (2)	6POD	Yes	Not anastomosed UV
	ROBOT	1	9:25	150	1 × UV, 2 × UOV	No	Temporary alopecia (1)	4POD	N/R	Not anastomosed UV
	ROBOT	2	10:48	100	1 × UV, 2 × UOV	N/R	Ureteral blood clot (3b)	6POD	N/R	
	ROBOT	3	12:10	200	1 × UV, 2 × UOV	N/R	Bilateral ureteral injury (3b)	3POD	N/R	Not anastomosed UV
	ROBOT	4	9:27	20	2 × UV, 2 × UOV	N/R	None	4POD	N/R	There were 2 left UOV
	ROBOT	5	12:03	100	3 × UOV	N/R	None	3POD	N/R	
Czech [7,25,26]	OPEN	1	5:20	100	2 × UV, 2 × OV	No	None	7POD	N/R	Not anastomosed UV
	OPEN	2	6:10	800	2 × UV, 2 × OV	No	None	7POD	N/R	Not anastomosed UV
	OPEN	3	7:10	100	2 × UV, 2 × OV	No	Climacteric symptoms (N/R)	6POD	N/R	Not anastomosed UV
	OPEN	4	5:30	100	2 × UV, 2 × OV	Yes	Bladder hypotonia (3a)	11POD	N/A	Not anastomosed OV
	OPEN	5	5:30	1000	2×UV, 2×OV	No	Ureter laceration (3a) Climacteric symptoms (N/R)	9POD	Yes	
Germany [27,28]	OPEN	1	12:07	100	2 × UV	No	None	11 days ^†^	Yes	
	OPEN	2	13:06	N/R	UV ^‡^	N/A	Hydronephrosis (3b)	N/R	N/A	No transplantation performed
	OPEN	3	9:03	100	1 × UV, 1 × OV	No	None	12 days^†^	Yes	
	OPEN	4	10:24	100	2 × UV, 1 × UOV	No	None	14 days ^†^	N/A	
	OPEN	5	9:11	100	2 × UV, 2 × UOV	No	None	14 days ^†^	N/A	
India [9,29]	LAP	1	4:00	100	1 × or 2 × UV, 2 × OV ^§^	No	None	7POD	N/R	
	LAP	2	4:00	100	1 × or 2 × UV, 2 × OV ^§^	No	None	7POD	N/R	
	LAP	3	2:40	100	2 × OV	No	None	6POD	N/R	
	LAP	4	3:20	100	2 × OV	No	None	6POD	N/R	

* Clavien-Dindo classification; † Hospital stay; ‡Number of removed UV not reported; ^§^ Used 1 x UV on No. 1 or No. 2 case. OPEN, open approach; N/R, not reported; UV, uterine vein; N/A, not applicable; UOV, utero-ovarian vein; POD, postoperative day; ROBOT, robot-assisted approach; OV, ovarian vein; UTI, urinary tract infection; LAP, laparoscopic approach.

**Table 2 jcm-10-00349-t002:** Operative and clinical data for each operative approach with or without using the uterine vein.

	OPEN			LAP			ROBOT		
	UV (+)	UV (−)	Total	UV (+)	UV (−)	Total	UV (+)	UV (−)	Total
*n*	29	4	33	2	2	4	12	2	14
Surgical time (h:min) *	8:45 ± 2:39	6:14 ± 0.26	8:26 ± 2:47	4:00 ± 0:00	3:00 ± 0.20	3:30 ± 0.33	11:19 ± 1:08	9:01 ± 3:01	10:59 ± 1:45
Blood loss (mL) *	711 ± 586	738 ± 569	715 ± 584	100 ± 0	100 ± 0	100 ± 0	245 ± 197	100 ± 0	209 ± 182
Discharge (POD)	6.3 ± 1.4	5.8 ± 0.8	6.2 ± 1.3	7.0 ± 0.0	6.0 ± 0.0	6.5 ± 0.5	4.4 ± 1.0	4.0 ± 0.0	4.3 ± 1.0
Complications (*n*,%)	17 (58.6%)	2 (50.0%)	19 (57.6%)	0 (0.0%)	0 (0.0%)	0 (0.0%)	6 (50.0%)	0 (0.0%)	6 (42.9%)
Graft failure (*n*,%)	8 (28.6%) ^†^	1 (25.0%)	9 (28.1%) ^†^	0 (0.0%)	0 (0.0%)	0 (0.0%)	2 (16.7%)	0 (0.0%)	2 (14.3%)
Live birth (*n*,%)	13 (46.4%)	3 (75.0%)	16 (48.5%)	0 (0.0%)	0 (0.0%)	0 (0.0%)	1 (8.3%)	1 (50.0%)	2 (14.3%)

* Mean ± SD; † Not including 1 case in which uterine transplantation was not performed. OPEN, open approach; LAP, laparoscopic approach; ROBOT, robot-assisted approach; UV, uterine vein; POD, postoperative day.

## Data Availability

Data available in a publicly accessible repository.

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
