# Peer review of "Operative and Clinical Outcomes of Minimally Invasive Living-Donor Surgery on Uterus Transplantation: A Literature Review"

_jcm, 2021, doi:10.3390/jcm10020349_

Round 1
Reviewer 1 Report
The review article "Operative and clinical outcomes of minimally invasive living-donor surgery on uterus transplantation: a literature review" performed by Matoba et al. including 26 English-written original articles from Saudi Arabia, Sweden, China, USA (Dallas), Czech Republic, Germany and India reports on the differences in surgical and clinical outcomes by the variation of surgical approach and the preserved veins in uterus transplantation living-donor surgery. The article`s core statements are that the minimally-invasive approaches may be introduced more frequently in living-donor uterus transplantation in the future and that the utilization of the ovarian veins or utero-ovarian veins instead of uterine veins for drainage may also contribute to less invasive living-donor surgery.
The living-donor uterus transplantation can be regarded as a very trendy issue in operative gynaecology since Professor Brännström from Sweden has achieved a live birth after transplanting a uterus from the living sister in a woman suffering from cervical cancer over six years ago.
The article is very well written and possesses a clear structure. Nevertheless, there are some small points which should be considered and revised.
-page 8, line 23: It should be "...tended to reduce blood loss..."
-page8, line 47: The sentence should be: Further accumulation of data which regards the clinical outcome of each approach is essential.
-In the introduction the authors also should mention an extended uterus myomatosus as a potential cause of absolute uterine factor infertility. There are regional differences, but especially in African countries it can be regarded as a serious problem.
Also the ethical dilemma of uterus transplantation with at least four operations (uterus explantation in the donor, uterus implantation in the infertile woman, caesarean section after pregnancy with the aid of assisted reproductive teschnologies, hysterectomy after completed family planning) should be discussed in a more detailed way. On the other Hand, infertility means a severe disease according to the WHO definition creating a large psychological burden, which may justify the surgical approaches.
All in all, in my opinion it is a very interesting article which can be published after a minor revision.
I am looking forward to reading your minor revision and I wish you the best for the new year 2021.
Author Response
Responses to the Reviewer
We are grateful to the reviewers and Editors for their critical comments and useful suggestions that have helped us to improve our paper considerably. As indicated in the responses that follow, we have taken all these comments and suggestions into account in the revised version of the paper.
Response to Reviewer #1
Comment 1: page 8, line 23: It should be "...tended to reduce blood loss..."
Answer: We apologize for this typographical error; this has been corrected in the manuscript.
Comment 2: page8, line 47: The sentence should be: Further accumulation of data which regards the clinical outcome of each approach is essential.
Answer: We have revised this description according to this comment.
Comment 3: In the introduction the authors also should mention an extended uterus myomatosis as a potential cause of absolute uterine factor infertility. There are regional differences, but especially in African countries it can be regarded as a serious problem.
Answer: We appreciate the reviewer’s suggestion. We have added “extended uterine myomatosis” according to the reviewer’s suggestion.
Comment 4: Also the ethical dilemma of uterus transplantation with at least four operations (uterus explantation in the donor, uterus implantation in the infertile woman, cesarean section after pregnancy with the aid of assisted reproductive technologies, hysterectomy after completed family planning) should be discussed in a more detailed way. On the other hand, infertility means a severe disease according to the WHO definition creating a large psychological burden, which may justify the surgical approaches.
Answer: As suggested, we have added these descriptions in the Discussion section.
Reviewer 2 Report
This paper gives an overview of the published cases of uterus transplantation in terms of the operative and clinical outcome. The report is well written in methods and result areas. The literature search is ok but unfortunately the discussion is speculative and it shows that the authors does not have experience of human uterus transplantation and the issues that should be discussed.
Major issues:
-The authors are stating that there is a gold standard for venous outflow in uterus transplantation. This is a bold statement. Since the procedure is so novel and with extremely few cases there is not yet a gold standard way of either procuring the organ nor transplanting it. Furthermore there has been no consensus by the teams performing the transplant in what a recommended venous outflow should be. There are a big variety in the veins used among cases and centers. Even in the Swedish study that the authors are referring to there were different combinations of veins used.
-Nomenclature: I would advise the authors to refer to the consensus report of nomenclature and reporting published by the USUTC (US uterus transplant consortium) in 2020 and use the same nomenclature. Utero-ovarian vein would then be changed to Superior uterine vein and the uterine vein should be referred to as the inferior uterine vein.
-The robotic approaches described from the different centers are not comparable without explanation. The Swedish cases were done only with a small portion with the robot (less than half of the given operative time and with the sensitive vessel dissection done openly) before they were converted to open surgery. The Dallas cases were done completely performed robotically with removal of the graft from the vagina.
-The authors state that:
"On the other hand, there are concerns about the use of the UOV and OV as drainage veins. The first is whether these veins are sufficient for blood flow in the gestational
uterus"
There are multiple reports on the sufficiency of using the superior uterine veins as only outflow after uterus transplant. Mainly by the Dallas team but also by other centers. Correct these statements and add references.
-The authors goes on with another ungrounded statement:
"In UTx, donor
vessels are mainly anastomosed to the recipient’s external iliac vein, but it is unclear how the uterine enlargement caused by pregnancy affects the anastomosis sites and vessels. In particular, when the UOV is used, the effect of uterine augmentation during pregnancy is likely to be significant because of the short length of vessels that can be removed. It is hoped that more data about these points will be collected."
There is no reports in the literature of the length not being sufficient of the superior uterine veins. On the contrary these vessels are almost always usable in terms of caliber and length (more so than the inferior uterine veins). The statement is purely speculative and not shared by centers performing uterus transplantation.
-The authors spend a lengthy discussion paragraph on the ovarian removal in premenopausal donors. I recommend removal of this section. It is very well known that removal of ovaries would lead to menopause and what symptoms are associated with menopause. I addition the authors state that "The use of the OV as a venous vessel on the premenopausal donor, where ovariectomy is inevitable, should be discussed".
Removal of ovaries should as a standard not be discussed with the premenopausal donor but avoided as stated by the ASRM guideline report for uterus transplantation.
Author Response
Responses to the Reviewer
We are grateful to the reviewers and Editors for their critical comments and useful suggestions that have helped us to improve our paper considerably. As indicated in the responses that follow, we have taken all these comments and suggestions into account in the revised version of the paper.
Response to Reviewer #ï¼’
Comment 1: The authors are stating that there is a gold standard for venous outflow in uterus transplantation. This is a bold statement. Since the procedure is so novel and with extremely few cases there is not yet a gold standard way of either procuring the organ nor transplanting it. Furthermore there has been no consensus by the teams performing the transplant in what a recommended venous outflow should be. There are a big variety in the veins used among cases and centers. Even in the Swedish study that the authors are referring to there were different combinations of veins used.
Answer: To the best of our knowledge, living donor surgery for uterine transplantation using uterine veins has been performed since the earliest days, and the number of procedures performed is high. However, the number of cases is still too small to make it the gold standard as the reviewer mentioned. We have changed the description so as not to cause misunderstanding.
Comment 2: Nomenclature: I would advise the authors to refer to the consensus report of nomenclature and reporting published by the USUTC (US uterus transplant consortium) in 2020 and use the same nomenclature. Utero-ovarian vein would then be changed to Superior uterine vein and the uterine vein should be referred to as the inferior uterine vein.
Answer: Although the consensus report of nomenclature and reporting published by the USUTC is available, we thought that those words are not used as internationally generalized yet. Therefore, in this review, I would like to use the terms “uterine vein”, “utero-ovarian vein”, and “ovarian vein”, which are used in many papers.
Comment 3: The robotic approaches described from the different centers are not comparable without explanation. The Swedish cases were done only with a small portion with the robot (less than half of the given operative time and with the sensitive vessel dissection done openly) before they were converted to open surgery. The Dallas cases were done completely performed robotically with removal of the graft from the vagina.
Answer: We have added these descriptions in the discussion section.
Comment 4: "On the other hand, there are concerns about the use of the UOV and OV as drainage veins. The first is whether these veins are sufficient for blood flow in the gestational uterus" There are multiple reports on the sufficiency of using the superior uterine veins as only outflow after uterus transplant. Mainly by the Dallas team but also by other centers. Correct these statements and add references.
Answer: We have already referred to the articles regarding successful births after UTx using OV or UOV only. Although we believe that the procedure with OV or UOV will be used more frequently in the future, the number of cases where the procedure was performed is still small. We believe that it is necessary to continue to examine whether OV or UOV is appropriate for the drainage vessels, and this is why it is described in this way.
Comment 5: The authors go on with another ungrounded statement: "In UTx, donor vessels are mainly anastomosed to the recipient’s external iliac vein, but it is unclear how the uterine enlargement caused by pregnancy affects the anastomosis sites and vessels. In particular, when the UOV is used, the effect of uterine augmentation during pregnancy is likely to be significant because of the short length of vessels that can be removed. It is hoped that more data about these points will be collected." There is no reports in the literature of the length not being sufficient of the superior uterine veins. On the contrary these vessels are almost always usable in terms of caliber and length (more so than the inferior uterine veins). The statement is purely speculative and not shared by centers performing uterus transplantation.
Answer: We have deleted these descriptions because it contains speculation as the reviewer points out.
Comment 6: The authors spend a lengthy discussion paragraph on the ovarian removal in premenopausal donors. I recommend removal of this section. It is very well known that removal of ovaries would lead to menopause and what symptoms are associated with menopause. I addition the authors state that "The use of the OV as a venous vessel on the premenopausal donor, where ovariectomy is inevitable, should be discussed". Removal of ovaries should as a standard not be discussed with the premenopausal donor but avoided as stated by the ASRM guideline report for uterus transplantation.
Answer: We have deleted these descriptions and added the ASRM statement on UTx according to the reviewer’s suggestion.